# Exogenous Melatonin Protects against Oxidative Damage to Membrane Lipids Caused by Some Sodium/Iodide Symporter Inhibitors in the Thyroid

**DOI:** 10.3390/antiox12091688

**Published:** 2023-08-29

**Authors:** Aleksandra K. Gładysz, Jan Stępniak, Małgorzata Karbownik-Lewińska

**Affiliations:** 1Department of Oncological Endocrinology, Medical University of Lodz, 7/9 Zeligowski St., 90-752 Lodz, Poland; aleksandra.gladysz@umed.lodz.pl (A.K.G.); jan.stepniak@umed.lodz.pl (J.S.); 2Polish Mother’s Memorial Hospital—Research Institute, 281/289 Rzgowska St., 93-338 Lodz, Poland

**Keywords:** melatonin, thyroid, sodium/iodide symporter inhibitors, lipid peroxidation

## Abstract

The thyroid gland is the primary site of sodium/iodide symporter (NIS), an intrinsic plasma membrane protein responsible for the active uptake of iodine, which is indispensable for thyroid hormone synthesis. Since exposure of the thyroid to NIS inhibitors can potentially have harmful effects on the entire organism, it is important to investigate the potential protective effects of known antioxidants, such as melatonin and indole-3-propionic acid (IPA), against pro-oxidative action of classic NIS inhibitors. The study aimed to check if and to what extent melatonin and IPA interact with some confirmed NIS inhibitors regarding their effects on oxidative damage to membrane lipids in the thyroid. For comparison with the thyroid gland, in which NIS is typically present, the liver tissue—not possessing NIS—was applied in the present study. Thyroid and liver homogenates were incubated in the presence of tested NIS inhibitors (i.e., NaClO_3_, NH_4_SCN, KSeCN, KNO_3_, NaF, KClO_4_, and BPA) in different ranges of concentrations with/without melatonin (5 mM) or IPA (5 mM). The malondialdehyde+4-hydroxyalkenals (MDA + 4-HDA) concentration (LPO index) was measured spectrophotometrically. NaClO_3_ increased LPO in the thyroid and in the liver, but these pro-oxidative effects were not prevented by either melatonin or IPA. Instead, pro-oxidative effects of NH_4_SCN observed in both tissues were prevented by both indole substances. KSeCN and NaF increased LPO only in the thyroid, and these pro-oxidative effects were prevented by melatonin and IPA. KNO_3_, KClO_4_, and BPA did not increase LPO, which can be due to their low concentrations resulting from restricted solubility. In conclusion, as melatonin prevented oxidative damage to membrane lipids in the thyroid caused by some sodium/iodide symporter inhibitors, this indoleamine shoud be considered as a potential protective agent when produced appropriately in living organisms but also as an exogenous substance recommended to individuals overexposed to NIS inhibitors.

## 1. Introduction

Sodium/iodide symporter (Na^+^/I^−^ symporter, NIS) is an intrinsic plasma membrane protein responsible for the active uptake of iodine from the blood into the thyroid gland, where it is required for thyroid hormone synthesis [1]. The expression of NIS has also been confirmed in other tissues, such as, for example, the salivary gland, the breast (during lactation), and the stomach [2]. Besides iodide, numerous other ions can interfere with the functions of NIS, such as perchlorate (ClO_4_^−^), thiocyanates (SCN^−^), and nitrate (NO_3_^−^) [3]. Therefore, these ions can consequently disrupt iodide uptake, which is the first and limiting step of thyroid hormone synthesis. Such substances that interfere with any aspect of hormone action are known as endocrine disruptors (also referred to as endocrine-disrupting chemicals or endocrine-disrupting compounds, EDCs). Whereas mechanisms of disrupting effects of EDCs are various, some of them can act via oxidative stress [4].

Exposure to EDCs primarily occurs through ingestion and, to a lesser extent, through inhalation and dermal uptake. In the case of NIS inhibitors, the primary route of exposure is through dietary intake of food. Substances like sodium chlorate (NaClO_3_) [5], ammonium thiocyanate (NH_4_SCN) [3], potassium nitrate (KNO_3_) [6], and potassium perchlorate (KClO_4_) [7] are frequently present in herbicides, defoliants, and fertilizers, and they can be readily absorbed by plants intended for consumption. Moreover, KNO_3_, in combination with salt, is used to preserve cured meats.

Regarding another route via drinking water, cases of water contamination involving excessive amounts of substances used for disinfection and water fluoridation have been documented [8,9]. Chlorine dioxide (ClO_2_), which has been increasingly used as a water disinfectant in recent years, effectively eliminates microorganisms without producing carcinogenic trihalomethanes and haloacetic acids. However, it should be noted that during the disinfection process with ClO_2_, harmful chlorates (ClO_3_^−^) are formed as a by-product [10]. Moreover, water fluoridation, a process still utilized by some countries to prevent and counteract tooth decay, can potentially be a source of harmful fluoride excess [11].

The main source of human exposure to SCN^−^ is the inhalation of cigarette smoke, as it contains cyanide (CN^−^) that can be easily metabolized into this ion [3].

Potential NIS inhibitors are also widely used in various industries, including NaClO_3_ in the paper and textile industry [12], KClO_4_ in explosives, fireworks, and rocket fuels [7], or NH_4_SCN in photography, anti-corrosion products, and laboratory analyses [3]. In turn, despite its confirmed harmfulness, bisphenol A (BPA), a flagship EDC, can still be found in food storage containers, plastic bottles, and other everyday items (e.g., toys and consumer electronics) [13].

While the mechanisms behind the disruptive effects of NIS inhibitors are diverse, some of them can act, similarly to other EDCs, through the induction of oxidative stress [4]. For example, the reduction of ClO_3_^−^ to chlorides generates reactive oxygen species (ROS) and free radicals that damage macromolecules [14,15]. Another instance is fluoride (F^−^), which induces oxidative stress by blocking electron transport in the mitochondria [16]. A similar mode of action has been observed in in vitro and in vivo studies with BPA [17,18] and SCN^−^ [19,20]. Previous studies have demonstrated that NO_3_^−^ ions in the body undergo transformation into nitrites, compounds that react with amines and amides to form N-nitroso compounds classified as carcinogenic and cytotoxic substances [21]. All these activities can potentially contribute to oxidative damage in tissues, such as the thyroid gland or the liver. Additionally, it should be stressed that thyroid hormone synthesis, which is initiated by an iodine uptake via NIS, comprises a series of chemical reactions associated with a significant production of free radicals and ROS [22,23]. On the other hand, selenium derivatives (SeCN^−^) have been shown in some studies to reduce oxidative stress by eliminating free radicals. Thus, their potential use can be considered in cancer prevention [24].

Antioxidants are compounds that scavenge and neutralize free radicals, thus reducing oxidative stress. Therefore, they can be considered agents potentially protective against oxidative damage caused by EDCs, such as NIS inhibitors. Some of the most effective antioxidants are indole substances.

Melatonin (N-acetyl-5-methoxytryptamine), the main metabolite of tryptophan, is the representative of the indole substances. Whereas the main physiological function of melatonin is the regulation of the circadian rhythm, it reveals numerous other actions in living organisms [25,26]. Of importance is the fact that melatonin is very powerful and effective in reducing oxidative damage to macromolecules due to numerous pathological conditions [27,28,29,30,31]. Other protective effects of melatonin discussed only recently in the literature comprise, among others, regulation of DNA methylation, which could contribute to cancer prevention [32], protection against vesicant warfare agents-induced inflammation and oxidative damage [33], or its favorable role in reproduction [34].

Another indole substance with a chemical structure similar to melatonin is indole-3-propionic acid (IPA), which also exhibits effective antioxidant properties. Whereas both indole substances, especially melatonin, have been shown to provide protection against oxidative damage to biological macromolecules caused by potential carcinogens [27,29,30,31,35], no studies have been conducted till now with respect to NIS inhibitors or generally EDCs.

This study aims to check if and to what extent melatonin and its related indole substances, i.e., indole-3-propionic acid, interact with some confirmed sodium/iodide symporter inhibitors regarding their effects on oxidative damage to membranes lipids (lipid peroxidation) in the thyroid. For comparison with the thyroid gland, in which sodium/iodide symporter is typically present, the liver tissue—not possessing sodium/iodide symporter—was applied in the present study.

It can be hypothesized that NIS inhibitors are able to induce oxidative damage in different tissues. However, it can occur more commonly in the thyroid, which is a tissue possessing NIS, the presence of which is associated with additional oxidative reactions [22,23]. Application with melatonin could probably alleviate, at least partially, the damaging effects of NIS inhibitors.

The potential application of current results may relate to health consequences when the human organism is exposed to high doses of NIS inhibitors, especially in occupational settings, and to possible protection caused by melatonin either administered as a supplement or coming from natural sources (endogenous production or a diet).

## 2. Materials and Methods

### 2.1. Ethical Considerations

In accordance with the Polish Act on the Protection of Animals Used for Scientific or Educational Purposes from 15 January 2015 (which implements Directive 2010/63/EU of the European Parliament and the Council of 22 September 2010 on the protection of animals used for scientific purposes)—the use of animals to collect organs or tissues does not require the approval of the Local Ethics Committee. These animals are only subjected to registration by the center in which the organs or tissues were taken. Additionally, we did not use experimental animals. Instead, porcine tissues were collected from animals at a slaughterhouse during the routine process of slaughter carried out for consumption.

### 2.2. Chemicals

Melatonin, indole-3-propionic acid (IPA), 17β-estradiol, sodium chlorate (NaClO_3_), ammonium thiocyanate (NH_4_SCN), potassium selenocyanate (KSeCN), potassium nitrate (KNO_3_), sodium fluoride (NaF), potassium perchlorate (KClO_4_) and bisphenol A (BPA) were purchased from Sigma (St. Louis, MO, USA). Ethanol (96%) was purchased from Stanlab (Lublin, Poland). The LPO-586 kit for LPO was obtained from Enzo Life Science (Farmingdale, NY, USA). All the used chemicals were of analytical grade and came from commercial sources.

### 2.3. Animals

Porcine thyroids and livers were collected from twenty-one (21) animals of both sexes (the animals were randomly chosen) at a slaughterhouse. All were sexually mature as determined by age (8–9 months) and body mass (119 ± 3.1 (SD) kg). After collection, the tissues were frozen on solid CO_2_ and stored at −80 °C until assay.

### 2.4. Experimental Steps

Thyroid and liver tissues were homogenized in ice-cold 50 mM Tris-HCl buffer (pH 7.4) (10%, *w*/*v*) and then incubated for 30 min at 37 °C in the presence of examined substances. Melatonin, IPA, 17β-estradiol, and BPA were dissolved in absolute ethanol. The concentration of ethanol in the final incubation medium was 1% (or 2% in the case of BPA + antioxidant) (*v*/*v*). Ethanol concentrations in controls were the same as in other samples in particular experiments. The NIS inhibitors (except BPA) were dissolved in Tris-HCl buffer.

In the preliminary experiments, we checked if available NIS inhibitors are able to induce LPO. Finally, in main experiments, co-incubations with chosen potential antioxidants were performed in thyroid homogenates in the case of all NIS inhibitors, but in liver homogenates only in the case of these NIS inhibitors which induced LPO:

1st experiment: NaClO_3_ (10.0, 7.5, 5.0, 2.5, 1.0, 0.75, 0.5, 0.25, 0.1, 0.075, and 0.05 mM) with or without melatonin (5 mM), IPA (5 mM), or 17β-estradiol (1 mM). Due to the lack of preventive effects of both indole substances, 17β-estradiol was used in this experiment.

2nd experiment: NH_4_SCN (1500, 1250, 1000, 750, 500, 250, 100, 75, 50, 25, and 10 mM) with or without melatonin (5 mM) or IPA (5 mM).

3rd experiment: KSeCN (500, 250, 100, 75, 50, 25, 10, 7.5, 5.0, 2.5, and 1.0 mM) with or without melatonin (5 mM) or IPA (5 mM).

4th experiment: KNO_3_ (10.0, 7.5, 5.0, 2.5, 1.0, 0.75, 0.5, 0.25, 0.1, 0.075, and 0.05 mM) and, additionally, in thyroid homogenates with or without melatonin (5 mM) or IPA (5 mM).

5th experiment: NaF (100, 75, 50, 25, 10, 7.5, 5.0, 2.5, 1.0, 0.75, and 0.5 mM) with or without melatonin (5 mM) or IPA (5 mM).

6th experiment: KClO_4_ (10.0, 7.5, 5.0, 2.5, 1.0, 0.75, 0.5, 0.25, 0.1, 0.075, and 0.05 mM) and, additionally, in thyroid homogenates with or without melatonin (5 mM) or IPA (5 mM).

7th experiment: BPA (2.0, 1.75, 1.5, 1.25, 1.0, 0.75, 0.5, 0.25, 0.1, 0.075, and 0.05 mM) and, additionally, in thyroid homogenates with or without melatonin (5 mM) or IPA (5 mM).

The highest achievable concentrations of NIS inhibitors, resulting from their limited solubility, were used in particular experiments. Melatonin concentration of 5 mM is the highest possible concentration resulting from its limited solubility in ethanol, and, therefore, it is commonly used in experimental studies [29,30,31,36,37,38]. Although IPA is characterized by a little bit higher solubility in ethanol, we decided to use the same concentration of IPA as that of melatonin.

Each experiment was run in duplicate and repeated three times.

### 2.5. Measurement of Lipid Peroxidation Products

The concentration of malondialdehyde + 4-hydroxyalkenals (MDA + 4-HDA), as an index of LPO, was measured in tissue homogenates, as described elsewhere [39]. Briefly, the homogenates were centrifuged at 3000× *g* for 10 min at 4 °C. After obtaining supernatant, each experiment was carried out in duplicates. The supernatant (200 µL) was mixed with 650 µL of a methanol:acetonitrile (1:3, *v*/*v*) solution containing a chromogenic reagent, N-methyl-2-phenylindole, and vortexed. Following the addition of 150 µL of methanesulfonic acid (15.4 M), the incubation was carried out at 45 °C for 40 min. The reaction between MDA + 4-HDA and N-methyl-2-phenylindole yields a chromophore, which was spectrophotometrically measured at the absorbance of 586 nm, using a solution of 10 mM 4-hydroxynonenal (4-HNE) as the standard. The level of lipid peroxidation was expressed as the amount of MDA + 4-HDA (nmol) per mg protein. Protein was measured using Bradford’s method [40], with bovine albumin as the standard.

Although this method is specifically calibrated to measure 4-HNE, it can also measure concentrations of other 4-hydroxyalkenals being homologous of 4-HNE, i.e., 4-hydroxydodeca-(2E,6Z)-dienal (4-HDDE) and also 4-hydroxy-2E-hexenal (4-HHE). The analytical limit of detection of this method is 0.5 nmol/mL.

### 2.6. Statistical Analyses

Data were statistically analyzed using a one-way analysis of variance (ANOVA), followed by the Student–Newman–Keuls test. Statistical significance was determined at the level of *p* < 0.05. Results are presented as the mean ± SE.

## 3. Results

In the 1st experiment, incubation of thyroid or liver homogenates in the presence of sodium chlorate (NaClO_3_) caused concentration-dependent increase in LPO levels, which were statistically significant for concentrations of 10.0, 7.5, 5.0, 2.5, 1.0, 0.75, and 0.5 mM in the thyroid or for concentrations of 10.0, 7.5, 5.0, 2.5, 1.0, 0.75, 0.5, and 0.25 mM in the liver. Melatonin (5 mM), indole-3-propionic acid (5 mM), and 17β-estradiol (1 mM) showed no protective effects either in the thyroid or in the liver. Unexpectedly, the addition of melatonin (5 mM) in combination with NaClO_3_ in its three highest concentrations (10.0, 7.5, 5.0 mM) resulted in a further increase in LPO levels only in the liver tissue (Figure 1A).

Baseline LPO level was higher in the liver than in the thyroid. LPO levels in response to NaClO_3_ (for all applied concentrations) were noticeably higher in the liver than in thyroid homogenates (Figure 1B).

In the 2nd experiment, incubation of homogenates in the presence of ammonium thiocyanate (NH_4_SCN) resulted in a statistically significant increase in the LPO level only for the middle two concentrations of 500 and 250 mM in the thyroid and for the middle three concentrations of 750, 500, and 250 mM in the liver. Both melatonin (5 mM) and indole-3-propionic acid (5 mM) completely prevented NH_4_SCN-induced LPO increases in both tissues (Figure 2A).

Baseline LPO level was higher in the liver than in the thyroid. NH_4_SCN-induced lipid peroxidation was higher in the liver than in thyroid homogenates, and that was observed for all concentrations of NH_4_SCN (except 1500 mM) (Figure 2B).

In the 3rd experiment, incubation of thyroid homogenates in the presence of potassium selenocyanate (KSeCN) resulted in a concentration-dependent increase (statistically significant for 500 mM) of LPO levels. Melatonin (5 mM) and indole-3-propionic acid (IPA) (5 mM) completely prevented KSeCN-induced LPO (Figure 3, top graph).

Incubation of liver homogenates in the presence of KSeCN did not change the level of LPO (Figure 3, bottom graph).

In the 4th experiment, incubation of thyroid or liver homogenates in the presence of potassium nitrate (KNO_3_) did not change the level of LPO (Figure 4).

In the 5th experiment, incubation of thyroid homogenates in the presence of sodium fluoride (NaF) resulted in a concentration-dependent increase (statistically significant for concentrations of 100, 75, 50, 25 mM) in LPO levels. Melatonin (5 mM) and indole-3-propionic acid (IPA) (5 mM) completely prevented NaF-induced LPO in thyroid homogenates (Figure 5, top graph).

Incubation of liver homogenates in the presence of NaF did not change the level of LPO (Figure 5, bottom graph).

In the 6th experiment, incubation of thyroid or liver homogenates in the presence of potassium perchlorate (KClO_4_) did not change the level of LPO (Figure 6).

In the 7th experiment, incubation of thyroid or liver homogenates in the presence of bisphenol A (BPA) did not change the level of LPO (Figure 7).

It should also be added that in all experiments, the following baseline LPO levels were lower in the thyroid vs. the liver: NaClO_3_—0.0779 vs. 0.122, *p* = 0.006; NH_4_SCN—0.0526 vs. 0.153, *p* = 0.002; KSeCN—0.102 vs. 0.131, *p* = 0.043; KNO_3_—0.0993 vs. 0.134, *p* = 0.078; NaF—0.0919 vs. 0.138, *p* = 0.002; KClO_4_—0.104 vs. 0.152, *p* = 0.002; BPA—0.0963 vs. 0.156, *p* = 0.022 (data shown in graphs only for the first two compounds, Figure 1B and Figure 2B). It should also be stressed that none of the antioxidants have changed the basal LPO level in the thyroid and—if examined—in the liver (Figure 1, Figure 2, Figure 3, Figure 4, Figure 5, Figure 6 and Figure 7). All results are summarized in Table 1.

## 4. Discussion

While both melatonin and IPA are repeatedly documented in the literature for their ability to prevent oxidative damage to macromolecules in various in vitro and in vivo models [27,28,29,30,31,35,41], our study is the first one to investigate the interactions of these indole substances with classic NIS inhibitors.

The ability of NIS to actively facilitate the uptake of I^−^ in the thyroid is crucial, as thyroid hormones play a vital role in the development and maturation of the central nervous system, skeletal muscle, and lungs in the regulation of intermediate metabolism in all cells, and they affect generally all body components throughout life. In addition to iodine, the NIS protein transports other ions, listed in order of decreasing intensity of transport: ClO_3_^−^, SCN^−^, SeCN^−^, NO_3_^−^, Br^−^, BF_4_^−^, IO_4_^−^, BrO_3_^−^, SO_4_^2−^, F^−^, HPO_4_^2−^, ReO_4_^−^ [42]. Regarding perchlorate (ClO_4_^−^), it is the strongest known NIS inhibitor [42], but it is not clear if it is transported by NIS inside the cell [43].

Because KClO_4_ in pharmacological doses blocks the active transport of iodine into the thyroid follicular cells and, subsequently, decreases thyroid hormone synthesis [3], it has been applied in the medical treatment of patients suffering from some types of hyperthyroidism [44].

It is remarkable that NIS exhibits the capability to transport a variety of substrates with different stoichiometric ratios. For instance, it transports I^−^, SCN^−^, and ClO_3_^−^ in an electrogenic stoichiometry of 2 Na^+^:1 anion. In contrast, NIS transports perrhenate (ReO_4_^−^) and perchlorate (ClO_4_^−^) in an electroneutral stoichiometry of 1 Na^+^:1 anion [43]. The transmembrane sodium gradient that enables iodine uptake is maintained by Na^+^/K^+^ ATPase [45] and can be experimentally inhibited by inhibitors typical of this enzyme, such as ouabain, thiocyanate (SCN^−^), or perchlorate (ClO_4_^−^) [46].

Due to the fact that NIS inhibitors, such as perchlorate, nitrate, and thiocyanate, can competitively interfere with iodine uptake by the thyroid gland, thereby affecting its function [3], exposure to these substances and similar compounds has been linked to developmental disorders. Previous studies have shown their neurotoxic effects [47,48,49,50], and prenatal exposure to EDC substances can impair prenatal growth, thyroid function, glucose metabolism, obesity, puberty, and fertility [51]. However, research regarding potential protection, among others by exogenous antioxidants, against detrimental effects of EDCs on humans is still limited. Examples of NIS inhibitors, of which the damaging effects are documented to be protected by melatonin, can be cadmium [52,53] or BPA [54]. Moreover, oxidative damage to membrane lipids in the thyroid caused by substantial excess of iodine compounds (the condition associated with inhibiting thyroid hormone synthesis) [36,37,38] or by potassium bromate (KBrO_3_) [30] has been documented by us to be prevented by melatonin or IPA.

As thyroid follicular cells are the primary site of NIS expression, and since exposure of the thyroid gland to NIS inhibitors can potentially cause deleterious effects on the entire organism, we have chosen this tissue to conduct our experiments. Additionally, it should be emphasized that the thyroid gland is an organ of “oxidative nature”, where oxidative processes are essential for hormone synthesis [22,23]. Therefore, any internal or external pathological factor that increases the oxidative load can disrupt the redox balance, leading to enhanced oxidative damage to macromolecules. In the current study, we also used the liver as a reference model of a tissue without NIS expression. However, this organ is also characterized by a high level of physiological oxidative burden [55]. Whereas melatonin is the main antioxidant examined in our study, we also used another indole substance structurally similar to melatonin and also with documented antioxidant properties, i.e., IPA.

In the present study, the following compounds were selected for analyses: sodium chlorate (NaClO_3_), ammonium thiocyanate (NH_4_SCN), potassium selenocyanate (KSeCN), potassium nitrate (KNO_3_), sodium fluoride (NaF), potassium perchlorate (KClO_4_), and bisphenol A (BPA). Unfortunately, ranges of concentrations of particular compounds differ substantially due to their different solubilities.

The first compound, i.e., NaClO_3_, characterized by one of the highest uptakes by NIS, increased lipid peroxidation in both the thyroid and the liver when used in a relatively low concentration of 10 mM. This observation suggests its very strong pro-oxidative properties, which, in turn, may explain why these damaging effects were not prevented by either melatonin or IPA. Therefore, due to the lack of preventive effects of indole substances, we checked if a well-known endogenous antioxidant 17β-estradiol was able to cause preventive action. Unfortunately, also 17β-estradiol applied in this model did not reveal any protection. It is worth mentioning that the antioxidative properties of 17β-estradiol have been confirmed, among others, in our earlier experimental studies [56,57,58]. However, this estrogen is not as effective as melatonin in protection against oxidative damage [36].

In the current study, melatonin, unexpectedly, enhanced the damaging effects of NaClO_3_ at its three highest concentrations of 10, 7.5, and 5 mM in the liver. Two possible reasons for this phenomenon can be considered. The first one may be associated with the potential pro-oxidative effect of melatonin, as suggested by some authors [59,60]. Namely, melatonin was found to promote ROS production when reducing a transient metal (Fenton reaction substrate) at low concentrations [59]. In addition, melatonin may interact with the mitochondria (probably mitochondrial electron transport chain complex III) to potentially promote ROS generation [60]. It is also worth noting that melatonin is a well-recognized pro-oxidant molecule in cancer cells, where it induces the production of ROS [61]. Thus, it is not excluded that pro-oxidative properties of melatonin may be also beneficial under other pathological conditions, such as overexposure to NIS inhibitors. However, it should be stressed again that evidence supporting the antioxidative action of this indoleamine prevail over that on pro-oxidative effects of melatonin. The other reason for the stimulatory effect of melatonin on NaClO_3_-induced oxidative damage in the liver may be an interaction with the method, resulting in a bias, especially considering relatively high concentrations of both compounds used. Thus, in our experimental model, no protective effects of melatonin on oxidative damage caused by NaClO_3_ were observed, either in the thyroid or in the liver. Therefore, our results do not provide any evidence that oxidative damage caused by this highly transportable ion via NIS can be prevented by potent antioxidants, at least under in vitro conditions.

Ammonium thiocyanate (NH_4_SCN) significantly increased LPO in both the thyroid and the liver, but only when this pro-oxidant was used in intermediate concentrations of 750, 500, and 250 mM. The lack of stimulatory effects observed at higher concentrations (1000–1500 mM) is similar to those observed in our earlier studies using potassium iodate (KIO_3_) [36,62]. The only possible explanation of this unexpected effect of NH_4_SCN could be that this pro-oxidant, being at very high concentrations, interacts with the method, resulting in a bias. It is of great importance to note that melatonin completely prevented oxidative damage to membrane lipids caused by NH_4_SCN, not only in the thyroid but also in the liver. Indole-3-propionic acid demonstrated a similar protective effect to that of melatonin in this model. In the context of current observation showing the protective action of indole substances against thiocyanate-induced oxidative damage, it should be recalled that treatment with melatonin reversed the enhanced oxidative damage to membrane lipids and improved skin biophysical characteristics in former smokers [63].

Potassium selenocyanate (KSeCN) increased lipid peroxidation only in the highest concentration of 500 mM and only in the thyroid. It is not excluded that the highest concentrations would be more damaging, but they were not achievable under in vitro conditions. Importantly, the increased LPO caused by KSeCN (500 mM) was completely prevented by melatonin and IPA.

The observation that potassium nitrate (KNO_3_) did not induce lipid peroxidation was rather expected, as its highest used concentration was 10 mM due to the low solubility of this compound. Importantly, neither melatonin nor IPA affected lipid peroxidation when used together with KNO_3_.

Sodium fluoride (NaF) caused a concentration-dependent increase in LPO, when used in the range of concentrations of 100–25 mM, but only in the thyroid, and this damaging effect was completely prevented by both indole substances.

The lack of a stimulatory effect of potassium perchlorate (ClO_4_^−^) on lipid peroxidation may be due to the low solubility of this compound (the highest achievable concentration of 10 mM). This phenomenon is similar to what was discussed above regarding potassium nitrate (NO_3_^−^). In our opinion, potassium perchlorate is this NIS inhibitor, which is especially worth examining in future studies due to its two exceptional features. The first one is that it is the strongest known NIS inhibitor [42]. The second one is that it has found application in the medical treatment of some types of hyperthyroidism [44]. Taking the above into account, looking for potential protective effects of melatonin against perchlorate-induced oxidative damage not only in the thyroid gland but in the entire organism would be of special value.

Bisphenol A was used in our study as a universal endocrine disruptor, although it is not documented to be a NIS inhibitor. Due to its very low solubility (only in ethanol), its highest achievable concentration was 2 mM. Expectedly, bisphenol A at the concentration of 2 mM did not change lipid peroxidation, and melatonin did not affect this process when used together with bisphenol A. Therefore, our experimental model did not allow us to compare the pro-oxidative effect of bisphenol A with those caused by NIS inhibitors or to evaluate the potential protective effect of melatonin against this classic EDC.

Summarizing our observation on pro-oxidative effects of particular NIS inhibitors, the following issues should be discussed. For KSeCN, KNO_3_, NaF, KClO_4_, and BPA, we did not detect any oxidative damage in the liver at all, whereas the lack of oxidative damage in the thyroid was found only in case of three pro-oxidants, i.e., KNO_3_, KClO_4_, and BPA. Therefore, the potential reasons should be discussed why KSeCN and, especially, NaF induced lipid peroxidation in the thyroid without any effects in the liver. It can be hypothesized that oxidative sensitivity of thyroid tissue to a higher number of NIS inhibitors is associated with the natural presence of NIS molecule in thyroidal tissue, which is able to transport iodine being an element of strong oxidative nature [4]. This feature of the thyroid together with the oxidative character of thyroid hormone synthesis [22,23] creates special oxidative conditions under which the response to pro-oxidants can differ from those in other tissues.

Regarding NaF, exposure to fluoride excess, especially combined with iodine deficiency, was found to have a negative impact on the brain and the thyroid, including oxidative damage to macromolecules, in both experimental and clinical studies [64]. In turn, why not only the thyroid but also the liver responded to pro-oxidative action of NaClO_3_ and NH_4_SCN? It should be stressed that these two compounds (NaClO_3_ and NH_4_SCN) possess the highest ability to be transported by NIS [42]. Together with the fact that the thyroid is a gland of an oxidative nature [22], it can be hypothesized that these two compounds are very strong pro-oxidants able to cause oxidative damage also in the liver. However, NaClO_3_ seems to be a much stronger pro-oxidative agent compared to NH_4_SCN as it increased LPO when used at a very low concentration of 10 mM, whereas NH_4_SCN was not effective even at the concentration one order of magnitude higher, i.e., 100 mM.

Regarding the effect of melatonin or IPA, it should be stressed that both antioxidants did not change the basal level of lipid peroxidation (shown in all experiments). This apparent lack of antioxidant effects of indole substances is expected and, in fact, favorable. As a certain level of oxidative stress is required in all cells and tissues under physiological conditions, diminishing the level of “physiological stress” is not required and could be even dangerous [22]. In agreement with our results, in other studies, no changes in lipid peroxidation were observed in response to melatonin only [29,30,31,36,37,38].

Potential mechanisms of protective effects of melatonin revealed in the present study against some NIS inhibitors should be discussed. They probably comprise receptor-independent scavenging of ROS and nitrogen species, such as superoxide anion radical (O_2_^•−^), hydrogen peroxide (H_2_O_2_), hydroxyl radical (•OH), nitric oxide (NO•), peroxynitrite anion (ONOO–), singlet oxygen (^1^O_2_), lipid peroxyl radical (LOO•), as well as receptor-dependent modulation of anti- and pro-oxidative enzymes, such as manganese superoxide dismutase (MnSOD), copper superoxide dismutase (CuSOD), glutathione peroxidase (GPx), glutathione reductase (GR), catalase (CAT), and gamma-glutamylcysteine synthase (γ-GC) [26]. It should be mentioned that metabolites of melatonin, such as cyclic 3-hydroxymelatonin (c3OHM), N-acetyl-N-formyl-5-methoxykynuramine (AFMK), or N-acetyl-5-methoxykynuramine (AMK) are also able to scavenge reactive species, sometimes more effectively than melatonin [26], but this mechanism may be of significance only in living organisms. With relation to the current study on NIS inhibitors, which are transported into thyroid follicular cells to different degree, it is crucial to recall that melatonin is able to cross all biological barriers and, therefore, it is present in all kinds of cells and in all subcellular compartments [26].

Although the protective effect of IPA was practically the same as that caused by melatonin, it should be emphasized that the former substance was used in our study only for comparison with melatonin. It should be stressed that melatonin, compared to IPA, is much better documented in the literature as an effective antioxidant [28], and, therefore, only its application in humans can be currently considered. Instead, IPA is a promising molecule regarding its clinical application, but more experimental and clinical evidence is still required. It is worth mentioning that in recent years, researchers have looked for even more effective derivatives of IPA concerning their protective antioxidative effects [65]. In the context of favorable actions of melatonin and IPA observed in the present study, our earlier finding should be mentioned, showing the cumulative protective effect of these two indole substances against experimentally induced oxidative damage to membrane lipids in the thyroid [37].

It is worth stressing that all NIS inhibitors used in our experiments, which did not increase LPO, were used in very low concentrations due to their restricted solubility. Therefore, it is not excluded that under in vivo conditions, these NIS inhibitors, when acting in higher concentrations, can induce oxidative damage to membrane lipids.

On the basis of current results, it is not possible to compare the damaging effects of particular NIS inhibitors on membrane lipids in the thyroid, as they were used at different concentration ranges. However, it seems that the degree to which the examined NIS inhibitors increased lipid peroxidation is not in agreement with the degree to which they are transported by NIS. Therefore, other chemical properties probably contributed substantially to pro-oxidative action of NIS inhibitors.

The last issue which should be discussed relates to the comparison between thyroid and liver tissues. The basal LPO level was somewhat higher in the liver than in the thyroid (which was observed in all our experiments), which is not unexpected as liver tissue is also characterized by high oxidative burden [55]. Whereas this difference in baseline LPO level in the liver and the thyroid is of interest and requires to be explored in future research, the broader discussion regarding this point is beyond the scope of the current manuscript. It can be only commented that higher baseline LPO levels in the liver vs. the thyroid could have contributed to higher lipid peroxidation induced by two NIS inhibitors, i.e., sodium chlorate (NaClO_3_) and ammonium thiocyanate (NH_4_SCN), in the liver vs. the thyroid. Therefore, the latter observation may not indicate any differences in sensitivity to a given pro-oxidant in examined tissues.

On the basis of our in vitro research, it would be interesting to design an in vivo study or even a clinical study. Regarding the first one, a given intact animal exposed to one or several NIS inhibitors will probably respond with increased oxidative damage in different tissues and organs, the thyroid gland included. Additionally, a blockade of NIS will probably occur with the transient inhibition of thyroid hormone synthesis, the phenomenon known as the Wolff–Chaikoff effect when it is caused by iodide excess [66,67]. In this context, it is worth mentioning that iodide-induced acute inhibition of NIS in animal thyroid follicular cells occurs via increased ROS production [68]. It is not excluded that NIS inhibitors, examined in the current study, also induce NIS blockade via ROS overproduction, which would further contribute to oxidative imbalance. As in some sensitive individuals exposed to iodide excess, inhibition of thyroid hormone synthesis may be chronic, leading to hypothyroidism [67], the same should be taken into account regarding overexposure to NIS inhibitors. In turn, co-exposure to high doses of melatonin or other indole substances would hopefully modify, at least partially, above unfavorable effects, i.e., it would diminish oxidative damage. However, there are no data in the literature on the effects of melatonin on iodine transportation. Regarding studies in humans, parameters of oxidative damage could be measured in body fluids, e.g., in urine and possibly in blood serum, collected from individuals exposed to particular NIS inhibitors, for example, in the workplace, who are additionally treated with melatonin in doses routinely recommended for biological rhythm regulation [28].

Regarding practical aspects of protective effects of melatonin observed in the present study, it should be considered what can be additional sources of this indoleamine which, under exposure to NIS inhibitors, would reveal preventive effects. Whereas appropriate exposure to light and darkness, especially avoidance of exposure to light at night [69], is the most physiological way to regulate melatonin synthesis, other natural sources, such as a diet, should also be considered. Research regarding this point has been rather poor till now, however, it has been shown that the administration of amino acids required for melatonin production, such as tryptophan, increased melatonin availability in rats [70]. Additionally, during the last decades, melatonin has been widely identified and qualified in various foods, from fungi to animals and plants [71]. It is well known that in different countries, exogenous melatonin is broadly used in tablets as a supplement. However, it should be stressed that exogenous melatonin is characterized by poor absolute bioavailability [72]. Additionally, although an appropriate amino acid administration and protein-rich diet may increase melatonin concentration in body fluids and tissues, these levels of melatonin are still much lower than those used in in vitro conditions. It can be recalled that melatonin concentration used in the present study (5 mM) exceeds by approximately three orders of magnitude the highest physiological concentrations of melatonin [73].

Our study certainly has some limitations. The first one relates to the fact that this is an in vitro study, therefore, our results cannot be directly extrapolated into in vivo conditions, especially regarding concentrations of used NIS inhibitors and of melatonin. It must be taken into account that under in vitro conditions, it is possible to examine an effect of a given single factor, whereas under in vivo conditions, such an effect can be modified by other numerous agents, sometimes completely reversing an initial action. The other limitation is that some NIS inhibitors were used only in low concentrations due to their poor solubility.

In summary, NIS inhibitors ((sodium chlorate (ClO_3_^−^), ammonium thiocyanate (SCN^−^), potassium selenocyanate (SeCN^−^) and sodium fluoride (F^−^)), when used in relatively high concentrations, are able to induce oxidative damage to membrane lipids in the thyroid gland, which is the tissue possessing NIS. Instead, in the liver, which is the tissue not expressing NIS, only two NIS inhibitors, namely sodium chlorate (ClO_3_^−^) and ammonium thiocyanate (SCN^−^), revealed their pro-oxidative effects. It should be stressed that among NIS inhibitors used in the present study, the most dangerous seems to be NaClO_3_ as it caused strong pro-oxidative effects when used in relatively low concentrations, and because these effects were not neutralized by indole substances and even melatonin enhanced its damaging effect in the liver. The most important finding of the present study is that melatonin and IPA completely prevented oxidative damage caused by ammonium thiocyanate (SCN^−^), potassium selenocyanate (SeCN^−^), and sodium fluoride (F^−^). Therefore, it should be taken into account that endogenous melatonin is able to prevent the detrimental effects of these three NIS inhibitors, which act as pro-oxidants. Additionally, the application of exogenous melatonin should be considered under conditions of high exposure to these agents in living organisms. The lack of a protective effect of melatonin against oxidative damage caused by sodium chlorate (ClO_3_^−^) in our in vitro study does not exclude its potential preventive action in living organisms. However, this assumption should be confirmed in in vivo experiments.

## 5. Conclusions

In conclusion, as melatonin prevented oxidative damage to membrane lipids in the thyroid caused by certain sodium/iodide symporter inhibitors, this indoleamine should be considered as a potential protective agent when produced appropriately in living organisms but also as an exogenous substance recommended to individuals overexposed to NIS inhibitors.

## Figures and Tables

**Figure 1 antioxidants-12-01688-f001:**
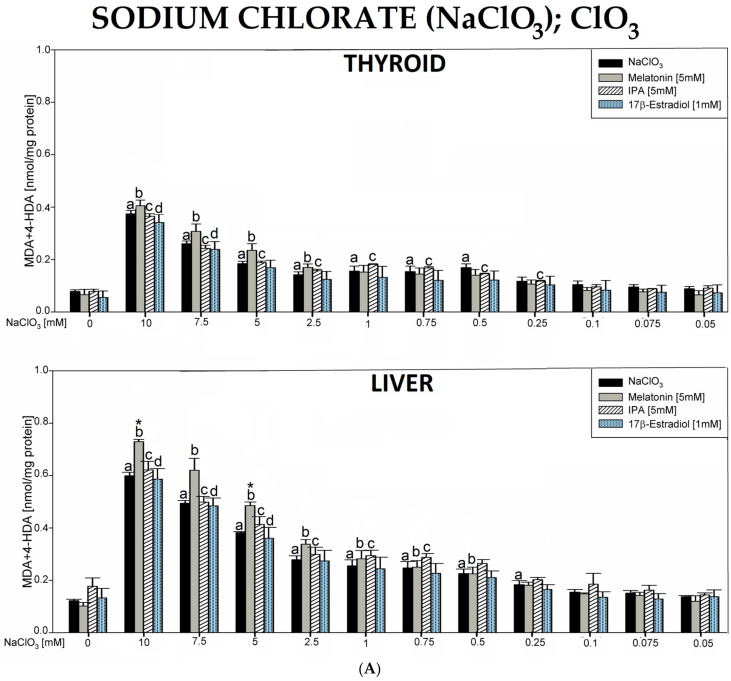
(**A**). Lipid peroxidation (LPO), measured as the level of malondialdehyde + 4-hydroxyalkenals (MDA + 4-HDA) in porcine thyroid or liver homogenates, incubated in the presence of sodium chlorate (NaClO_3_) (10.0, 7.5, 5.0, 2.5, 1.0, 0.75, 0.5, 0.25, 0.1, 0.075, and 0.05 mM) alone (black bars), or NaClO_3_ with melatonin (5 mM) (grey bars) or NaClO_3_ with indole-3-propionic acid (IPA) (5 mM) (striped bars) or NaClO_3_ with 17β-estradiol (1 mM) (dot bars). LPO level is expressed in nmol/mg protein. Data are presented as mean ± SE (error bars). ^a^
*p* < 0.05 vs. control (without any substance); ^b^
*p* < 0.05 vs. melatonin (5 mM); ^c^
*p* < 0.05 vs. indole-3-propionic acid (5 mM); ^d^
*p* < 0.05 vs. 17β-estradiol (1 mM); * *p* < 0.05 vs. NaClO_3_ in the same concentration. (**B**). Lipid peroxidation (LPO), measured as the level of malondialdehyde + 4-hydroxyalkenals (MDA + 4-HDA), in porcine thyroid (black bars) or liver (grey bars) homogenates. Homogenates were incubated in the presence of sodium chloride (NaClO_3_) (10.0, 7.5, 5.0, 2.5, 1.0, 0.75, 0.5, 0.25, 0.1, 0.075, and 0.05 mM) used to induce lipid peroxidation. LPO level is expressed in nmol/mg protein. Data are presented as mean ± SE (error bars). ^a^
*p* < 0.05 vs. control in the thyroid; ^b^
*p* < 0.05 vs. control in the liver; * *p* < 0.05 vs. NaClO_3_ in the same concentration in the other tissue.

**Figure 2 antioxidants-12-01688-f002:**
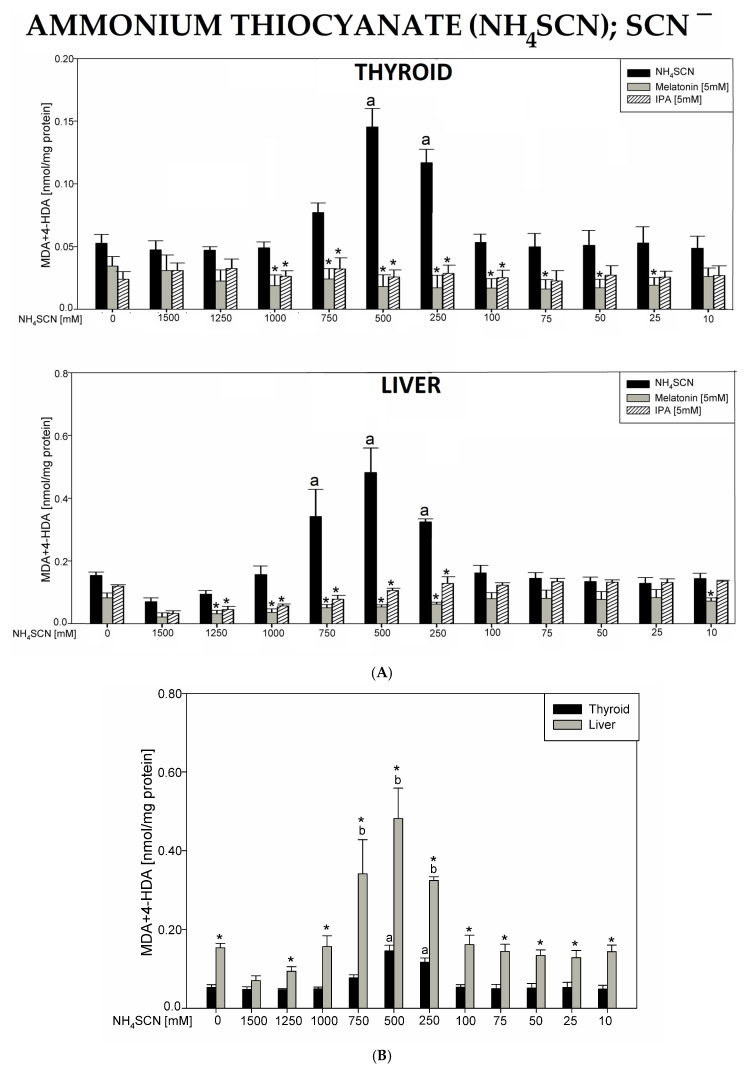
(**A**). Lipid peroxidation (LPO), measured as the level of malondialdehyde + 4-hydroxyalkenals (MDA + 4-HDA) in porcine thyroid or liver homogenates, incubated in the presence of ammonium thiocyanate (NH_4_SCN) (1500, 1250, 1000, 750, 500, 250, 100, 75, 50, 25, and 10 mM) alone (black bars), or NH_4_SCN plus melatonin (5 mM) (grey bars) or NH_4_SCN plus indole-3-propionic acid (IPA) (5 mM) (striped bars). LPO level is expressed in nmol/mg protein. Data are presented as mean ± SE (error bars). ^a^
*p* < 0.05 vs. control (without any substance); * *p* < 0.05 vs. NH_4_SCN in the same concentration. (**B**). Lipid peroxidation (LPO), measured as the level of malondialdehyde + 4-hydroxyalkenals (MDA + 4-HDA) in porcine thyroid (black bars) or liver (grey bars) homogenates. Homogenates were incubated in the presence of ammonium thiocyanate (NH_4_SCN) (1500, 1250, 1000, 750, 500, 250, 100, 75, 50, 25, and 10 mM) used to induce lipid peroxidation. LPO level is expressed in nmol/mg protein. Data are presented as mean ± SE (error bars). ^a^
*p* < 0.05 vs. control in the thyroid; ^b^
*p* < 0.05 vs. control in the liver; * *p* < 0.05 vs. NH_4_SCN in the same concentration in the other tissue.

**Figure 3 antioxidants-12-01688-f003:**
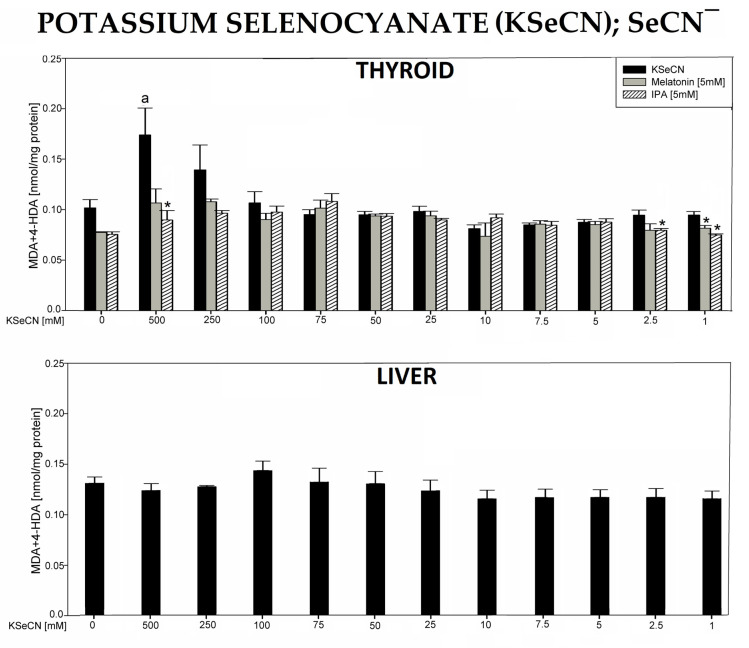
Lipid peroxidation (LPO), measured as the level of malondialdehyde + 4-hydroxyalkenals (MDA + 4-HDA) in porcine thyroid or liver homogenates, incubated in the presence of potassium selenocyanate (KSeCN) (500, 250, 100, 75, 50, 25, 10, 7.5, 5.0, 2.5, and 1.0 mM) alone (black bars), or KSeCN plus melatonin (5 mM) (grey bars) or KSeCN plus indole-3-propionic acid (IPA) (5 mM) (striped bars). LPO level is expressed in nmol/mg protein. Data are presented as mean ± SE (error bars). ^a^
*p* < 0.05 vs. control (without any substance); * *p* < 0.05 vs. KSeCN in the same concentration.

**Figure 4 antioxidants-12-01688-f004:**
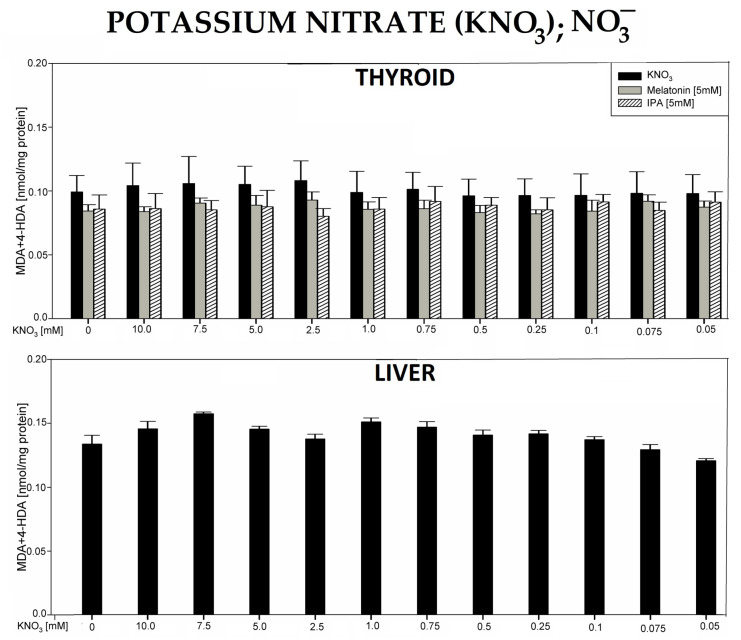
Lipid peroxidation (LPO), measured as the level of malondialdehyde + 4-hydroxyalkenals (MDA + 4-HDA) in porcine thyroid or liver homogenates, incubated in the presence of potassium nitrate (KNO_3_) (10.0, 7.5, 5.0, 2.5, 1.0, 0.75, 0.5, 0.25, 0.1 0.075, and 0.05 mM) or—in case of thyroid tissue—KNO_3_ plus melatonin (5 mM) (grey bars) or KNO_3_ plus indole-3-propionic acid (IPA) (5 mM) (striped bars). LPO level is expressed in nmol/mg protein. Data are presented as mean ± SE (error bars). No significant differences were found.

**Figure 5 antioxidants-12-01688-f005:**
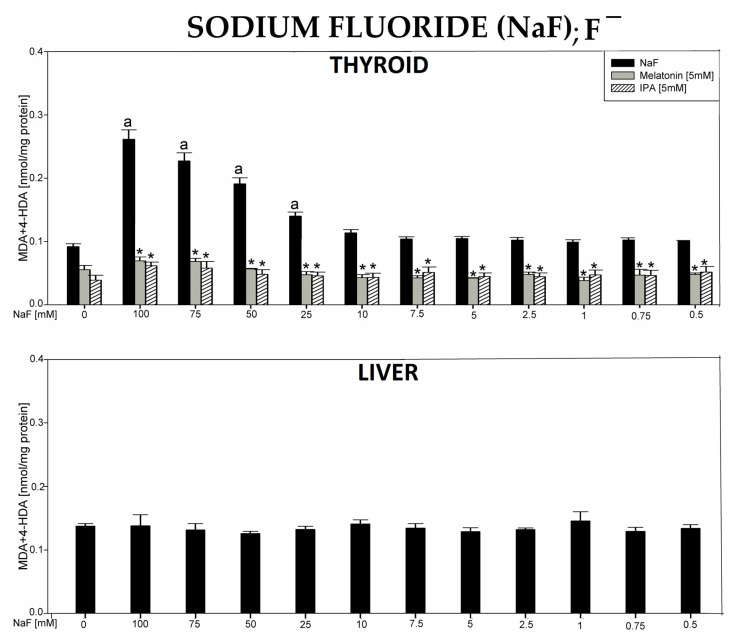
Lipid peroxidation (LPO), measured as the level of malondialdehyde + 4-hydroxyalkenals (MDA + 4-HDA) in porcine thyroid or liver homogenates, incubated in the presence of sodium fluoride (NaF) (100, 75, 50, 25, 10, 7.5, 5.0, 2.5, 1.0, 0.75, and 0.5 mM) alone (black bars), or—in case of thyroid tissue—NaF plus melatonin (5 mM) (grey bars) or NaF plus indole-3-propionic acid (IPA) (5 mM) (striped bars). LPO level is expressed in nmol/mg protein. Data are presented as mean ± SE (error bars). ^a^
*p* < 0.05 vs. control (without any substance); * *p* < 0.05 vs. NaF in the same concentration.

**Figure 6 antioxidants-12-01688-f006:**
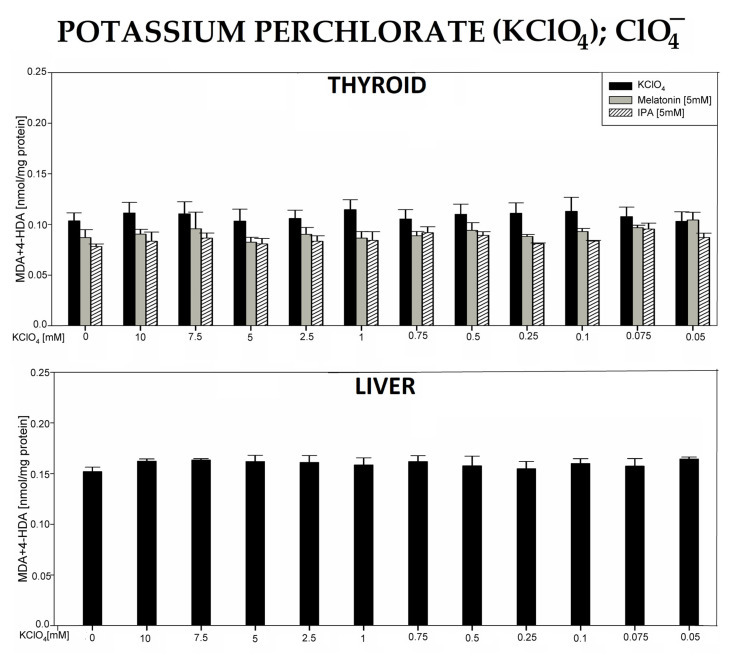
Lipid peroxidation (LPO), measured as the level of malondialdehyde + 4-hydroxyalkenals (MDA + 4-HDA), in porcine thyroid or liver homogenates, incubated in the presence of potassium perchlorate (KClO_4_) (10.0, 7.5, 5.0, 2.5, 1.0, 0.75, 0.5, 0.25, 0.1, 0.075, and 0.05 mM) or—in case of thyroid tissue—KClO_4_ plus melatonin (5 mM) (grey bars) or KClO_4_ plus indole-3-propionic acid (IPA) (5 mM) (striped bars). LPO level is expressed in nmol/mg protein. Data are presented as mean ± SE (error bars).

**Figure 7 antioxidants-12-01688-f007:**
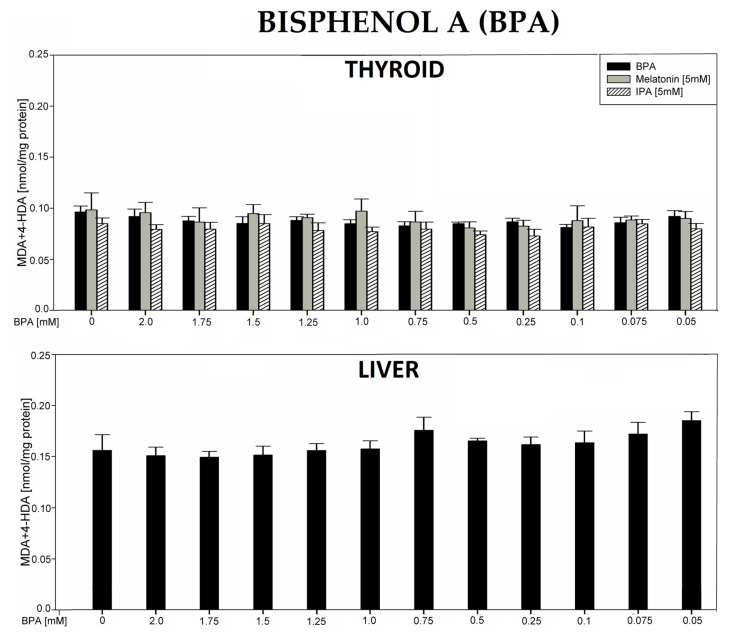
Lipid peroxidation (LPO), measured as the level of malondialdehyde + 4-hydroxyalkenals (MDA + 4-HDA) in porcine thyroid or liver homogenates, incubated in the presence of bisphenol A (BPA) (2.0, 1.75, 1.5, 1.25, 1.0, 0.75, 0.5, 0.25, 0.1, 0.075, and 0.05 mM) alone (black bars) or—in case of thyroid tissue—BPA plus melatonin (5 mM) (grey bars) or BPA plus indole-3-propionic acid (IPA) (5 mM) (striped bars). LPO level is expressed in nmol/mg protein. Data are presented as mean ± SE (error bars).

**Table 1 antioxidants-12-01688-t001:** Schematic summary of results presented in Figure 1, Figure 2, Figure 3, Figure 4, Figure 5, Figure 6 and Figure 7. ↑, the increased lipid peroxidation (LPO); (+), protective effect caused by melatonin or indole-3-propionic acid (IPA); (−), no effect.

NIS Inhibitors (Ion, a Full Compound) the Highest Used Concentration	Lipid Peroxidation (LPO)		Protection by Melatonin/IPA	
	Thyroid	Liver	Thyroid	Liver
ClO_3_^−^ Sodium chlorate (NaClO_3_) 10 mM	↑	↑	−	−
SCN^−^ Ammonium thiocyanate (NH_4_SCN) 1500 mM (but 750–250 mM effective)	↑	↑	+	+
SeCN^−^ Potassium selenocyanate (KSeCN) 500 mM	↑	−	+	−
NO_3_^−^ Potassium nitrate (KNO_3_) 10 mM	−	−	−	−
F^−^ Sodium fluoride (NaF) 100 mM	↑	−	+	−
ClO_4_^−^ Potassium perchlorate (KClO_4_) 10 mM	−	−	−	−
Bisphenol A (BPA) 2.0 mM	−	−	−	−

## Data Availability

The datasets used and/or analyzed during the current study are available from the corresponding author on reasonable request.

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
