# Peer review of "Exogenous Melatonin Protects against Oxidative Damage to Membrane Lipids Caused by Some Sodium/Iodide Symporter Inhibitors in the Thyroid"

_antioxidants, 2023, doi:10.3390/antiox12091688_

Round 1

Reviewer 1 Report

Title: 

Exogenous melatonin protects against oxidative damage to membrane lipids caused by sodium/iodide symporter inhibitors in the thyroid.

Authors: 

Gladysz A.K., Stepniak J., Karbownik-Lewinska M.

Manuscript ID: antioxidants-2532576

Objective: 

In this manuscript, the antioxidant effect of melatonin and indole-3-propionic acid (IPA) as protection for sodium/iodide symporter (NIS) at intrinsic plasma membrane protein (responsible for iodine uptake into the thyroid) was tested. For this purpose, 7 NIS inhibitors, i.e., NaClO3, NH4SCN, KSeCN, KNO3, NaF, KClO4 and BPA, each with 11 different concentrations. The authors concluded that melatonin and IPA can prevent the oxidative stress caused by NIS inhibitors.  

Points of criticism:

Isolated typing errors, e.g., page 2 - line 86 or page 10 – line 295 should be improved.

The authors mention only the thyroid gland in the title, although the melatonin effect also applies to the liver. In this respect, the title should be adapted.

On page 4 - line 161, a method for lipid peroxidation products, i.e., MDA + 4-HDA, is mentioned. This parameter is essential, primarily since the entire study design is based on its results. Although a citation is given, it is necessary to describe the method briefly. In particular, it would be interesting to know which 4-hydroxyalkenals are measured and the sensitivity or standard values. 

Page 4 – line 176:

The authors found increased stress values for NaClO3 at the concentrations (10, 7.5, and 5 mM). This prooxidant effect of Melatonin should be emphasized not only by referring to references 59 and 60 but in the form of an attempt at explanation – including a bias through the method (Line 361), and the physiologically relevant concentrations for NaClOshould be indicated, e.g., in a table for each NIS inhibitor. Furthermore, it should be emphasized that NaClO3 could not be neutralized by the antioxidants (indoleamine) and is thus most dangerous for NIS. Why did the authors use 17?-estradiol and then only in one experiment?

Another peculiarity was found for NH4SCN (page 5 – line 201), which gave the highest OS values in both tissues in the middle concentrations (750, 500 and 250 mM). This item should be discussed with possible explanations, including normal ranges.

Figure 2a shows significant baseline level differences between the thyroid and liver, although the levels are similar in almost all other experimental approaches. The authors should explain this divergence.

Figures 1a, 2a and 5 are too small, and the bars are too close together, which makes it difficult to recognize significant information. A more precise presentation should be made here.

For KSeCN, KNO3, NaF, KClO4 and BPA, the authors did not detect any oxidative stress in the liver at all. This detail should be explained and discussed compared to the thyroid gland about KSeCN and NaF.

In this context, the increased baseline values in the liver with BPA must also be discussed. 

How did the authors determine the antioxidant concentration of melatonin and IPA? On the one hand, this concentration seems effective against increased oxidative stress caused by NIS inhibitors. Still, there is no significant improvement by these antioxidants at the zero value. This should be discussed.

The authors note (line 436 ff) - that the results cannot be readily transferred to in vivo conditions about NIS inhibitors and melatonin without indicating this as a weakness. Furthermore, some substances were only used in low concentrations due to poor water solubility. This should also be indicated as a weakness of the study, and a table with corresponding standard values or cut-off values should be included.

Author Response

Points of criticism:

Isolated typing errors, e.g., page 2 - line 86 or page 10 – line 295 should be improved.

We would like to thank very much the Reviewer for this remark. Therefore, we have made appropriate changes, i.e. the word  “consider” we have changed to “considered’ (p.2, line 89) and we have removed unnecessary space (p.14, line 340). All other typing errors which occurred throughout the text are also removed.

The authors mention only the thyroid gland in the title, although the melatonin effect also applies to the liver. In this respect, the title should be adapted.

We would like to thank very much the Reviewer for this suggestion. Initially we have also considered such a conception, however, protective effect of melatonin/IPA in the liver was observed in case of only one compound, i.e. ammonium thiocyanate (NH4SCN), whereas in case of the thyroid – protection caused by melatonin/IPA was observed in case of three (of four) compounds; therefore we wrote “caused by some…inhibitors” in the title. Additionally, the liver was used in the present study as a tissue not possessing NIS just for comparison with effects observed in the thyroid as a typical tissue possessing NIS. Taking all above into account we decided not to change the title in the current revision. However, if the Reviewer still suggests to do that, we are ready to make appropriate changes in the title in the next revision.

On page 4 - line 161, a method for lipid peroxidation products, i.e., MDA + 4-HDA, is mentioned. This parameter is essential, primarily since the entire study design is based on its results. Although a citation is given, it is necessary to describe the method briefly. In particular, it would be interesting to know which 4-hydroxyalkenals are measured and the sensitivity or standard values.

We would like to thank very much the Reviewer for this remark and we absolutely agree with the suggestion that the description of the method should be more detailed. Therefore we have added an appropriate text, also regarding 4-hydroxyalkenals and the sensitivity of the method (p. 4-5; lines 181-195).

Page 4 – line 176:

The authors found increased stress values for NaClO3 at the concentrations (10, 7.5, and 5 mM). This prooxidant effect of Melatonin should be emphasized not only by referring to references 59 and 60 but in the form of an attempt at explanation – including a bias through the method (Line 361), and the physiologically relevant concentrations for NaClO3 should be indicated, e.g., in a table for each NIS inhibitor. Furthermore, it should be emphasized that NaClO3 could not be neutralized by the antioxidants (indoleamine) and is thus most dangerous for NIS. Why did the authors use 17?-estradiol and then only in one experiment?

We would like to thank very much the Reviewer for this remark. We have found the mistake in the description of the results therefore we have corrected it (p. 5; lines 204,209).

Whereas the Reviewer suggests to discuss obtained results in relation to “physiologically relevant concentrations for NaClO3”, it is impossible to find such concentrations in the literature; the same concerns all other NIS inhibitors. Therefore we do not have a basis to prepare an additional table with such values. Regarding lethal doses and maximum tolerated doses of some NIS inhibitors they can be found in the literature, however, they do not define final concentrations in living organisms and, after all, these details would be beyond the scope of the current manuscript. However, if the Reviewer still requires we can add the values of lethal doses or maximum tolerated doses in the next revision.

Concerning most dangerous effects of NaClO3, observed in the current study, we have added appropriate text in the Discussion section (p. 19, lines 591-595).

17β-estradiol was used in this experiment due to the lack of preventive effects of both indole substances and this applies only to the first study with the use of NaClO3. Therefore, we have added an appropriate sentence in Materials and Methods section (p. 4; lines 158-159). Additionally, we have extended the discussion on this point by adding an appropriate text (p. 15; lines 390-393).

Another peculiarity was found for NH4SCN (page 5 – line 201), which gave the highest OS values in both tissues in the middle concentrations (750, 500 and 250 mM). This item should be discussed with possible explanations, including normal ranges.

We would like to thank very much the Reviewer for this remark. We added a short explanation (p. 16, lines 422-424); however, it is difficult to explain these unexpected results taking account normal ranges, as such reference ranges for NH4SCN are not specified in the literature. At this moment, the added explanation is the only which we are able to propose. However, if the Reviewer is not satisfied with this explanation we will try to rethink and rewrite this discussion in the next revision. In our previous studies [63], regarding KIO3 effect (iodate induced LPO in the middle concentrations) we were able to discuss this unexpected results with relation to normal ranges of iodine because physiological concentrations of iodine in the thyroid tissue are just specified in the literature.

Figure 2a shows significant baseline level differences between the thyroid and liver, although the levels are similar in almost all other experimental approaches. The authors should explain this divergence.

We would like to thank very much the Reviewer for this remark. Now we realized that we did not specify statistically significant differences between the thyroid and the liver regarding LPO levels in particular experiments. In fact statistically significant differences were found in case of all experiments performed in the present study but the differences have been shown in graphs only regarding two first NIS inhibitors i.e. in Figures 1B and 2B. Because it is not justified to draw Figures regarding other five NIS inhibitors to compare LPO levels between the thyroid and the liver without antioxidants, we prepared an additional text containing LPO levels (and p-values) in the thyroid and the liver under basal conditions in particular experiments and we have added this text into Results section (p. 5, line 211; p. 6, line 239; p. 13, lines 318-324). Additionally, we have modified the Discussion text regarding this issue (p. 18, lines 530-539).

Figures 1a, 2a and 5 are too small, and the bars are too close together, which makes it difficult to recognize significant information. A more precise presentation should be made here.

We would like to thank very much the Reviewer for this remark. As the Reviewer suggested we have modified the figures.

For KSeCN, KNO3, NaF, KClO4 and BPA, the authors did not detect any oxidative stress in the liver at all. This detail should be explained and discussed compared to the thyroid gland about KSeCN and NaF.

We would like to thank very the much Reviewer for this remark. Therefore we have added an appropriate paragraph in the Discussion section (p. 17, lines 460-483).

In this context, the increased baseline values in the liver with BPA must also be discussed.

This issue has already been discussed above and appropriate text has been added into the manuscript (p. 5, line 211; p. 6, line 239; p. 13, lines 318-324 and p. 18, lines 530-539). Actually, it is not important that the experiment was carried out with BPA or any other NIS inhibitor, because baseline LPO (thus without addition of any substance) was measured under the same conditions in each experiment.

How did the authors determine the antioxidant concentration of melatonin and IPA? On the one hand, this concentration seems effective against increased oxidative stress caused by NIS inhibitors. Still, there is no significant improvement by these antioxidants at the zero value. This should be discussed.

We would like to thank very much the Reviewer for this remark and we have added an adequate text (p. 4, lines 173-177), also in the Discussion section (p. 17, lines 484-490).

The authors note (line 436 ff) - that the results cannot be readily transferred to in vivo conditions about NIS inhibitors and melatonin without indicating this as a weakness. Furthermore, some substances were only used in low concentrations due to poor water solubility. This should also be indicated as a weakness of the study, and a table with corresponding standard values or cut-off values should be included.

We would like to thank very much the Reviewer for this remark. Therefore we have added a paragraph on limitations of the study (p. 19, lines 578-585). Regarding a table with corresponding standard values or cut-off values, we decided at this moment not to prepare because standard values are not clearly described in the literature and regarding cut-off values this may be available only regarding tolerable doses which are not directly translated to tissue concentrations. We think that such data are beyond the scope of our manuscript which after all is very long and contains many details. However, if the Reviewer still requires we will add suggested data in the next revision.

Reviewer 2 Report

The manuscript of Gladysz et al. aims to evaluate the possible positive effects of two major anti-oxidant molecules (melatonin and indole-3-propionic acid (IPA)), on the potential oxidative stress elicited by the inhibitors of the Sodium/Iodide symporter (NIS). To achieve this goal, Authors performed a series of in vitro experiments to 1) evaluate the malondialdehyde+4-hydroxyalkenals (MDA+4-HDA) concentration (LPO index) as a read-out of oxidative damage in a NIS-expressing tissue (thyroid) compared to a tissue lacking NIS (liver) upon incubation with diverse NIS inhibitors; 2) measure the protective, anti-oxidant effects on LPO index of melatonin and IPA in these tissue homogenates. Authors finds that Sodium chlorate (NaClO3) and Ammonium thyocyanate (NH4SCN) both induced LPO in thyroid and in liver, but only the effect of NH4SCN was diminished by melatonin/IPA. The LPO-inducing effect of Potassium selenocyanate (KSeCN) and Sodium fluoride (NaF) was specific for the thyroid and the augmented LPO by both agents was attenuated by melatonin/IPA. Taken their data together, authors provide compelling evidence of the complexity of potential harmful effects of commonly found environmental chemicals on thyroid function and the function of other metabolic organs. 

The manuscript is accompanied by 7 multipanel Figures and 1 Summary Table. Authors cite 65 references to put their findings in context.   The manuscript is easy-to-follow, the Figure Legends provide the necessary information to interpret the graphs.

The topic fits the scope of the journal “Antioxidants” though it is of moderate interest due to its limited scope (only in vitro) experimental set-up and the narrow focus of investigations restricted only to the measurement of LPO as a read-out for oxidative stress. On the positive side however, Authors clearly deliberate this limitation in the “Discussion” part. Altogether, this is a descriptive paper better fitted for other MDPI journals (e.g. Toxics).

Authors should address the following issues before the manuscript could be considered for acceptation in any journal:

1.     The data of the manuscript do not correspond to the title; in fact, there were only 65% of tested NIS transporter inhibitors whose effect was blocked by melatonin and some of them was not specific for the thyroid. Authors should change this misleading 

2.     It is not clear from the “Methods” part whether matching amount of ethanol was included in the control samples where the effects of antioxidants (melatonin/BPA, both dissolved in ethanol) were examined. 

Author Response

  1. The data of the manuscript do not correspond to the title; in fact, there were only 65% of tested NIS transporter inhibitors whose effect was blocked by melatonin and some of them was not specific for the thyroid. Authors should change this misleading 

We would like to thank very much the Reviewer for this suggestion. We have changed the title to the following “Exogenous melatonin protects against oxidative damage to membrane lipids caused by some sodium/iodide symporter inhibitors in the thyroid”. This is the fact that melatonin is not specific to the thyroid, however, in the other tissue examined, i.e. in the liver, melatonin was protective only in case of one NIS inhibitor. Additionally, the liver was used in the present study as a tissue not possessing NIS just for comparison with effects observed in the thyroid as a typical tissue possessing NIS. However, if the Reviewer still suggests to put data on liver tissue into the title, we are ready to make appropriate changes in the next revision.

  1. It is not clear from the “Methods” part whether matching amount of ethanol was included in the control samples where the effects of antioxidants (melatonin/BPA, both dissolved in ethanol) were examined.

We would like to thank very much the Reviewer for this remark. We absolutely agree that this issue should be specified in the manuscript therefore we have added an appropriate sentence [p. 4, lines 150-151]

Reviewer 3 Report

The manuscript submitted to Antioxidants by Gladysz et al., titled: "Exogenous melatonin protects against oxidative damage to membrane lipids caused by sodium/iodide symporter inhibitors in the thyroid" is an interesting ex vivo study aiming to investigate the effects of exogenously administered melatonin on oxidative damage to membranes induced by Na/I supporter inhibitors in the thyroid. 

The reviewer would like to present some points below for the authors' consideration. 

1. What is the rationale for the aim of the study? Please consider making it clear also drawing the connection to potential application (clinical or public health).

2. Consider including a hypothesis statement in the introduction section.

3. While this is described as an ex vivo study it would be interesting to consider including the number of animals and the description of age, weight, sex as these are factors of potential variation in terms of the vitality and physiological response characteristics of the cells.

4. Consider including a rationale for the determination of the doses in terms of all critical compounds (melatonin, inhibitors etc).

5. An interesting point of discussion would be to work on the paradigm of either an in vivo study or a clinical study. How would an intact animal challenged with oxidative stress and/or thyroid pathology respond to such a regime? Along similar lines what could be predicted for humans?

6. Could regulation through either amino acid administration or protein rich diet be achieved in terms of melatonin availability similarly to the effects proposed by the study herein? This again would be an interesting point of discussion looking at the topic from an applied sciences perspective. 

There are parts of the manuscript where English is somewhat challenging to follow. The authors tend to write long sentences without the appropriate syntax and grammar quality. Their manuscript should be reviewed by a native English speaker for flow and syntax. Also there is a number of typos which would need to be addressed.

Author Response

The reviewer would like to present some points below for the authors' consideration.

  1. What is the rationale for the aim of the study? Please consider making it clear also drawing the connection to potential application (clinical or public health).

We would like to thank very much the Reviewer for this remark. Therefore we have added the appropriate text (p. 3, lines118-121).

  1. Consider including a hypothesis statement in the introduction section.

We would like to thank very much the Reviewer for this remark. Therefore we have added the appropriate text (p. 3, lines113-117)

  1. While this is described as an ex vivo study it would be interesting to consider including the number of animals and the description of age, weight, sex as these are factors of potential variation in terms of the vitality and physiological response characteristics of the cells.

We would like to thank very much the Reviewer for this remark. Therefore we have added the appropriate text (p. 3-4, lines 141-144).

  1. Consider including a rationale for the determination of the doses in terms of all critical compounds (melatonin, inhibitors etc).

We would like to thank very much the Reviewer for this remark. Therefore, we have added the appropriate text (p. 4, lines 173-177).

  1. An interesting point of discussion would be to work on the paradigm of either an in vivo study or a clinical study. How would an intact animal challenged with oxidative stress and/or thyroid pathology respond to such a regime? Along similar lines what could be predicted for humans?

We would like to thank very much the Reviewer for this remark. Therefore, we have added a paragraph in the Discussion section (p. 18 line 540-559). 

  1. Could regulation through either amino acid administration or protein rich diet be achieved in terms of melatonin availability similarly to the effects proposed by the study herein? This again would be an interesting point of discussion looking at the topic from an applied sciences perspective.

We would like to thank very much the Reviewer for this remark. Therefore, we have added a short discussion regarding this issue (p. 18-19, lines 560-577).

Comments on the Quality of English Language

There are parts of the manuscript where English is somewhat challenging to follow. The authors tend to write long sentences without the appropriate syntax and grammar quality. Their manuscript should be reviewed by a native English speaker for flow and syntax. Also there is a number of typos which would need to be addressed.

We would like to thank very much the Reviewer for this remark. We have done our best to improve the language of the manuscript. We hope that now the text is more comprehensive.

Round 2

Reviewer 1 Report

The authors have responded to all questions and have made changes and improvements to the manuscript where possible. In all other cases, the relevant reasons were given and discussed. Thus, I thank you for the professional author response and wish you all the best for the publication.

Reviewer 2 Report

Authors responded to the points raised by this reviewer.

English language is acceptable.

Reviewer 3 Report

The authors have made a reasonable effort in addressing reviewer's comments. Proofreading is suggested.